# Assessment of the Microbial Spoilage and Quality of Marinated Chicken Souvlaki through Spectroscopic and Biomimetic Sensors and Data Fusion

**DOI:** 10.3390/microorganisms10112251

**Published:** 2022-11-14

**Authors:** Evgenia D. Spyrelli, George-John E. Nychas, Efstathios Z. Panagou

**Affiliations:** Laboratory of Microbiology and Biotechnology of Foods, Department of Food Science and Human Nutrition, School of Food and Nutritional Sciences, Agricultural University of Athens, Iera Odos 75, 11855 Athens, Greece

**Keywords:** poultry, spectroscopic methods, microbiological quality, biomimetic sensor, multivariate data analysis, data fusion

## Abstract

Fourier-transform infrared spectroscopy (FT-IR), multispectral imaging (MSI), and an electronic nose (E-nose) were implemented individually and in combination in an attempt to investigate and, hence, identify the complexity of the phenomenon of spoilage in poultry. For this purpose, marinated chicken souvlaki samples were subjected to storage experiments (isothermal conditions: 0, 5, and 10 °C; dynamic temperature conditions: 12 h at 0 °C, 8 h at 5 °C, and 4 h at 10 °C) under aerobic conditions. At pre-determined intervals, samples were microbiologically analyzed for the enumeration of total viable counts (TVCs) and *Pseudomonas* spp., while, in parallel, FT-IR, MSI, and E-nose measurements were acquired. Quantitative models of partial least squares–Regression (PLS-R) and support vector machine–regression (SVM-R) (separately for each sensor and in combination) were developed and validated for the estimation of TVCs in marinated chicken souvlaki. Furthermore, classification models of linear discriminant analysis (LDA), linear support vector machine (LSVM), and cubic support vector machines (CSVM) that classified samples into two quality classes (non-spoiled or spoiled) were optimized and evaluated. The model performance was assessed with data obtained by six different analysts and three different batches of marinated souvlaki. Concerning the estimation of the TVCs via the PLS-R model, the most efficient prediction was obtained with spectral data from MSI (root mean squared error—RMSE: 0.998 log CFU/g), as well as with combined data from FT-IR/MSI (RMSE: 0.983 log CFU/g). From the developed SVM-R models, the predictions derived from MSI and FT-IR/MSI data accurately estimated the TVCs with RMSE values of 0.973 and 0.999 log CFU/g, respectively. For the two-class models, the combined data from the FT-IR/MSI instruments analyzed with the CSVM algorithm provided an overall accuracy of 87.5%, followed by the MSI spectral data analyzed with LSVM, with an overall accuracy of 80%. The abovementioned findings highlighted the efficacy of these non-invasive rapid methods when used individually and in combination for the assessment of spoilage in marinated chicken products regardless of the impact of the analyst, season, or batch.

## 1. Introduction

Spectroscopic methods, such as Fourier-transform infrared spectroscopy (FT-IR) and multispectral Imaging (MSI), have been investigated in tandem with regression and classification algorithms for their effectiveness in the assessment of the quality of meat and poultry [1,2,3]. FT-IR vibrational spectroscopy has been recommended as an efficient solution for the discrimination of intact chicken breast muscle during spoilage via partial least squares–discriminant analysis (PLS-DA) and outer product analysis (OPA) [4]. PLS-R coupled with FT-IR successfully detected the microbial loads of chicken breast [5,6]. Furthermore, FT-IR analysis was proven to be an appropriate tool for the identification of chicken meat among other types of raw food through PLS-R and support vector machine (SVM) classification based on FT-IR data [7]. Regarding MSI analysis, it has been proposed as a reliable method in tandem with a PLS-R model for the estimation of microbial groups associated with the spoilage of chicken meat [8,9,10]. Likewise, MSI analysis and the implementation of PLS-R accurately predicted the time from slaughter in four poultry products [11]. Nevertheless, this nondestructive method has been suggested as an alternative for the detection of food fraud in minced pork adulterated with chicken [12]. Both MSI and FTIR analysis have been investigated for their ability to feasibly predict TVCs and *Pseudomonas* spp. on the surface of stored chicken thigh fillets, and they could accurately classify chicken samples into two quality classes [13].

Another important indicator related to microbiological spoilage in food is the volatile profile associated with the metabolic activity of the microbiota [14]. An electronic nose (E-nose) is a biomimetic technology that is modeled after the olfactory system of humans, and it comprises an array of electronic chemical sensors that record odors via volatiles [14,15,16]. The main advantage of this method over spectroscopic methods is the low number of derived results, which is more convenient for multivariate data analysis due to the reduced noise in the dataset [17,18]. This environmentally friendly approach has been examined for its efficacy in assessing quality and microbial spoilage in red meat and poultry by using a variety of regression and discrimination models [19,20,21]. An E-nose and a PLS-R implementation was proposed for the estimation of chicken fat [16,22]. Moreover, an E-nose was successfully employed in combination with an SVM-R model for the prediction of TVCs in chilled pork [23] and in the indigenous microbiota of beef fillets [24]. Apart from the development of SVM models, E-nose signals have been used in the development of back-propagation neural networks (BPNNs) for the prediction of TVCs in chicken [25], as well as in the implementation of a variety of machine learning models for the determination of microbial groups in minced meat [26]. In addition, E-nose data that were analyzed with the LDA and BP-ANN models were evaluated for their potential to detect pork freshness via the volatile colorific fingerprint obtained during the spoilage of pork samples [27].

However, each of these rapid and non-invasive methods has its own advantages, weaknesses, and limitations concerning the monitoring and controlling procedures for quality and safety in the meat industry [17]. Taking into consideration the complexity of the food matrix during meat spoilage in terms of physical, biological, and chemical properties, a combination of sensor features could capture both internal (metabolites, chemical compounds) and external (color, smell, texture, tenderness) alterations more effectively and, thus, identify quality defects in food more accurately [20,28]. In this context, the fusion of data from different sensors was recently investigated for its synergistic role in the improvement of model classification and/or prediction potential [29]. For meat products, low and mid-fusion have been employed as two different data merging techniques for the development of models that predict quality, freshness, microbial loads [28,30], and adulteration [31]. Classification models generated with E-nose, computer vision (CV), and artificial tactile (AT) data demonstrated accurate predictions of pork and chicken freshness [32]. An ensemble of spectral, textural, and color features was proven efficient via a classification model of k-mean-BFF for the assessment of quality in chicken meat [33], whereas the combination of an E-nose (colorimetric sensors array) and hyperspectral imaging successfully estimated chicken meat quality and freshness [34]. In addition, fusion of data from two spectral methods—namely, V-NIR and SWIR—was suggested as a feasible solution for the tracing of foreign materials (FMs) on the surface of chicken breast fillets [35]. 

The aim of this study was the development of quantitative and qualitative models for rapidly assessing spoilage in marinated chicken souvlaki via MSI, FT-IR, and E-nose measurements when used both individually and in combination (mid-fusion). PLS-R and SVM-R models were developed for the determination of TVCs in marinated chicken souvlaki. Further on, LDA, LSVM, and CSVM classification models were developed on sensor data (both individually and in combination) for the detection of two quality classes. A model performance assessment with data from independent batches and analysts confirmed the efficacy of nondestructive techniques and their feasibility for being performed even by untrained personnel.

## 2. Materials and Methods

### 2.1. Experimental Design

Marinated chicken souvlaki (n = 209, *ca* 48.89 ± 1.3 g) samples were transferred from a Greek poultry industry to the laboratory (within 24 h from the slaughter and marinating process), placed in Styrofoam trays (two portions per tray), and wrapped with clingfilm. After packaging, samples were stored aerobically in three isothermal conditions, namely 0, 5, and 10 °C (two independent experiments), and one dynamic temperature profile (12 h at 0 °C, 8 h at 5 °C, and 4 h at 10 °C) in high-precision (±0.5 °C) incubation chambers (MIR-153, Sanyo Electric Co., Osaka, Japan), where the temperature was monitored every 20 min by data loggers (CoxTracer, Belmont, NC, USA). At predetermined intervals, the samples were microbiologically analyzed (enumeration of TVCs and *Pseudomonas* spp.) while, simultaneously, FT-IR, MSI, and E-nose data were acquired. At each sampling point, duplicate samples stored at the same isothermal condition (n = 2 × 2) and triplicate samples stored at the dynamic temperature profile (n = 3) were subjected to the above-mentioned analyses. The microbiological results were expressed as log CFU/g. Then, quantitative and qualitative models that assessed microbial spoilage and quality in the marinated chicken souvlaki were developed and validated. PLS-R and SVM-R models were employed separately for the estimation of TVCs for each sensor. Moreover, mid-level fusion (preprocessing of data via principal component analysis (PCA) and then the employment of PLS-R) was performed for the evaluation of the combined use of MSI, FT-IR, and E-nose sensors for the assessment of TVCs. In the same context, classification models that used linear discriminant analysis (LDA), a linear support vector machine (LSMV), and a cubic support vector machine (CSVM) were evaluated for their efficacy in identifying 2 spoilage classes via the MSI, FT-IR, and E-nose data (in combination and separately).

For validation, storage experiments under aerobic isothermal conditions (0, 4, 5, 8, and 10 °C) were undertaken by different analysts (n = 6) with three different batches of marinated chicken souvlaki. MSI, FT-IR, and E-nose measurements were collected and correlated with the respective TVC results. The quantitative and qualitative models that were developed were fitted to the experimental data obtained in order to evaluate their performance.

### 2.2. Microbiological Analysis

A portion of 25 g of marinated chicken souvlaki (chicken thigh fillet, sodium chloride, sodium acetate, sodium citrate, enzyme tenderizer, and ascorbic acid) was transferred aseptically to a Stomacher bag containing 225 mL of sterile quarter-strength Ringer’s solution (Lab M Limited, Lancashire, UK) and was homogenized by a Stomacher device (Lab Blender 400, Seward Medical, UK) for 60 s. From this 1:10 sample solution, serial decimal dilutions were prepared using the same diluent, and 0.1 mL of the appropriate dilution was spread to the following media: (a) tryptic glucose yeast agar (Plate Count Agar, Biolife, Milan, Italy) for the enumeration of total viable counts (TVCs), when incubated at 25 °C for 72 h; (b) Pseudomonas agar base (LAB108 supplemented with selective supplementation with Cetrimide Fucidin Cephaloridine, Modified C.F.C. X108, LABM) for the determination of the presumptive *Pseudomonas* spp. counts, when incubated at 25 °C for 48 h.

### 2.3. Sensors

#### 2.3.1. Spectral Acquisition

The marinated chicken souvlaki samples were subjected to MSI analysis by using the Videometer-Lab instrument (Videometer A/S, Herlev, Denmark), which captured the surface reflectance of the samples from 18 monochromatic wavelengths (405–970 nm), namely, 405, 435, 450, 470, 505, 525, 570, 590, 630, 645, 660, 700, 850, 870, 890, 910, 940, and 970 nm. The description of this sensor and the process of image acquisition have been thoroughly discussed in previous studies [36]. The final outcome of this acquisition was a data cube of size m × n × 18 (where m × n is the image size in pixels) containing spatial and spectral data for each sample [37]. Prior to data analysis, an extra step was needed in which the region of interest (ROI) on the sample surface was separated from the surrounding area, which contained non-useful information. For each image, the mean reflectance spectrum was estimated with the calculation of the average value and the standard deviation of the intensity of pixels within the ROI at each wavelength. For this purpose, canonical discriminant analysis (CDA) was applied individually to each sample and for each wavelength by using the Videometer-Lab version 2.12.39 software (Videometer A/S, Herlev, Denmark).

FT-IR analysis was implemented using a ZnSe 45 HATR (Horizontal Attenuated Total Reflectance) crystal (PIKE Technologies, Madison, WI, USA) and an FT-IR-6200 JASCO spectrometer (Jasco Corp., Tokyo, Japan). The ATR crystal showed a refractive index of 2.4 and a depth of penetration of 2.0 μm at 1000 cm^−1^. Spectra were obtained in the wavenumber range of 4000 to 400 cm^−1^ using the Spectra Manager Code of Federal Regulations (CFR) software version 2 (Jasco Corp., Tokyo, Japan); for this, 100 scans with a resolution of 4 cm^−1^ and a total integration time of 2 min were accumulated. Background measurements were undertaken [38] and the absorption was subtracted via the software of the instrument.

#### 2.3.2. Electronic Nose (E-Nose)

An Alpha M.O.S a-FOX sensor array system 3000 (Alpha M.O.S, Toulouse, France) with 12 metal oxide sensors was used in this study. The system consisted of a sampling apparatus, an array of sensors, air-generating equipment (F-DGSi, Evri, France), and software (Alpha Soft V12.46) for data recording. The sensor array contained 12 metal oxide sensors that were divided into the T, P, and LY types; these were LY2/LG, LY2/G, LY2/AA, LY2/GH, LY2/gGTL, LY2/gGT, T30/1, P10/1, P10/2, P 40/1, T 70/2, and PA/2 [39]. Prior to injection, 2 g of the sample (2.013 ± 0.002 g) was placed in a 2.5 mL vial, sealed with aluminum caps, and heated at 50 °C for 20 min in a thermoblock 2t static-headspace sampler (Teknokroma Analitica S.A., Barcelona, Spain). A volume of 500 μL of the generated headspace was injected into the E-nose with an injection rate of 500 μL/s. The method parameters were defined as follows: (a) acquisition duration: 120 s; (b) acquisition period: 1 s; (c) acquisition time: 800 s; (d) gas flow (air): 150 mL/min. The signal response of each array was expressed in the form of relative resistance changes (Delta R/Ro).

### 2.4. Data Processing

The MSI spectral data consisted of 18 mean values and the 18 respective standard deviations of the intensity in pixels for each observation/measurement. The spectral data were preprocessed with a standard normal variance (SNV) transformation to remove collinear and “noisy” data [40]. The same transformation was applied to the E-nose data, which contained the relative resistance for each sensor. For the FT-IR spectral data, the Savitzky–Golay second-derivative transformation (second-order polynomial, second derivative, nine-point window) was applied on the spectra at wavelengths in the range of 900 to 2000 cm^−1^ for the reduction of the baseline shift and noise [41].

Both qualitative and quantitative models were calibrated and optimized (k-fold cross-validation, k = 10) with data from the storage experiments at 0, 5, and 10 °C and the dynamic temperature profile (n = 169), while an external validation of the model was performed with independent data (n = 40, 3 batches, 6 analysts) from the storage experiments under aerobic isothermal (0, 4, 5, 8, 10 °C) conditions. For the development of the classification models based on two quality classes, the samples were defined as non-spoiled (class 1: 42.01%) and spoiled (class 2: 57.99%) when the TVCs were below or above 7.0 log CFU/g, respectively. The developed two-class classification models were evaluated for their performance with data from independent experiments (n = 40; class 1: 22 samples (55%); class 2: 18 samples (45%)).

### 2.5. Model Development and Performance Assessment

The development of the PLS-R models for each sensor individually and in combination was performed by using the Unscrambler© ver. 9.7 software (CAMO Software AS, Oslo, Norway). For the single-sensor models, the acquired sensor data were correlated with TVCs via the development of the PLS-R models. Principal component analysis (PCA) was applied separately to the sensor data, and the derived PCA scores were merged for the development of the two-sensor and three-sensor PLS-R models [42]. Similarly, support vector machines–regression (SVM-R) was employed for the estimation of the TVCs by using the MATLAB 2012a software (The MathWorks, Inc., Natick, MA, USA), where the single-sensor, two-sensor, and three-sensor models were performed and validated. The performance of the developed PLS-R and SVM-R models was evaluated using the correlation coefficient (r) and the root mean squared error (RMSE: log CFU/g) indices. Furthermore, classification models using LDA, LSVM, and CSVM were developed via the MATLAB 2012a software and assessed for their accuracy using the following performance metrics: overall accuracy (%), sensitivity (%), and precision (%) [43].

## 3. Results

### 3.1. Microbiological Results

The population dynamics of TVCs and *Pseudomonas* spp. during aerobic storage under isothermal conditions and the dynamic temperature profile are presented in Figure 1 and Figure 2**,** respectively. The initial TVCs and *Pseudomonas* spp. counts in the marinated chicken samples stored in isothermal conditions were enumerated at 5.37 (±0.26) and 5.01 (±0.01) log CFU/g, respectively, whereas these microbial groups reached 5.26 (±0.06) and 4.43 (±0.10) log CFU/g in samples from the dynamic temperature profile. The storage temperature had a great impact on the TVCs and behavior of *Pseudomonas* spp. in the samples, with the spoilage of the chicken occurring at different time points. Specifically, the TVCs reached the threshold of spoilage (7.0 log CFU/g) [44] at 0 °C in 216 h (7.41 ± 0.74 log CFU/g), at 5 °C in 72 h (7.07 ± 0.73 log CFU/g), and at 10 °C in 42 h (7.01 ± 0.6 log CFU/g). The population of TVCs in the samples maintained with the dynamic temperature profile followed a similar growth behavior to that of the samples stored at 5 °C, as their counts reached 6.89 (±0.42) log CFU/g in 96 h. Likewise, the *Pseudomonas* spp. counts, which are associated with the production of slime and off-odors when they reach 7.0 log CFU/g in meat products [45,46], reached this limit at 0 °C in 216 h (7.06 ± 1.04 log CFU/g), at 5 °C in 96 h (6.86 ± 0.76 log CFU/g), at 10 °C in 48 h (7.61 ± 0.38 log CFU/g), and with the dynamic temperature profile in 120 h (6.46 ± 0.81 log CFU/g).

### 3.2. Spectra and E-Nose Signals

The E-nose signal response (intensity) for each sensor is shown in Figure 3A for a non-spoiled (0 h) and a spoiled (240 h) sample of marinated chicken souvlaki stored at 5 °C. As shown in Figure 3B, differences in the intensity between non-spoiled and spoiled samples also occurred for six sensors, namely, PA/2, T30/1, P10/1, P10/2, P40/1, and T70/2. The first sensor was linked with changes in ethanol, ammonia, and organic amines [39], which were due to the proteolytic activity of *Pseudomonas* spp. during meat spoilage [47]. The results from the P40/1 sensor were related to the presence of fluorine [23,48]. The remaining sensors—T30/1, P10/1, P10/2, and T70/2—could be associated with organic solvents, hydrocarbons, methane, and aromatic compounds, respectively [48], and subsequently with the formation of biofilm by *Pseudomonas* spp. during meat spoilage [23,49].

The FT-IR and MSI spectra for non-spoiled (0 h at 0 °C) and spoiled (240 h at 5 °C) marinated chicken souvlaki are presented in Figure 4. Regarding the MSI spectra (Figure 4B), the reflectance (mean intensity in pixels) in the non-spoiled and spoiled samples differed at 660, 700, 850, 870, 890, 910, and 940 nm; the region of 660 to 700 nm is related to the myoglobin in meat color, as described elsewhere [11]. From the FT-IR results (Figure 4A), absorption bands showing variations between the non-spoiled and spoiled samples were located in the areas of 1000.87–1150 cm^−1^ and 1476.24–1692.2 cm^−1^. Specifically, absorption bands at 1541.81 and 1629.55 cm^−1^ were attributed to metabolic products (amide I and II) that were associated with the microorganisms involved in spoilage, such as *Pseudomonas* spp., and their metabolic activity on the surface of meat during spoilage [4,50]. 

### 3.3. PLS-R Models for Assessing TVC Loads in Marinated Chicken Souvlaki

The PLS-R models’ parameters (slope, offset) and performance metrics (r, RMSE) are presented in Table 1 for model optimization (full cross-validation) and evaluation (prediction) for each sensor separately and in combination. The MSI model exhibited the highest performance during prediction, with an RMSE value of 0.998 log CFU/g, which was within the acceptable microbial prediction zone of ±1.0 log CFU/g. Likewise, the FT-IR model showed an RMSE of the prediction values of 1.025 log CFU/g. On the contrary, the E-nose model poorly predicted the TVCs, with an RMSE of the prediction value of 1.921 log CFU/g. Of the combined two-sensor models, the combination of FT-IR/MSI outperformed all of the others (RMSE: 0.983 log CFU/g), followed by the MSI/E-nose sensor. The combined three-sensor model demonstrated an RMSE value of 1.367 during prediction, while for the E-nose/FT-IR model, this value was 1.757 log CFU/g. Additionally, the values of the correlation coefficient (r) ranged from 0.722 to 0.803, except for the E-nose, which presented a very low value for this performance index (r = 0.245). The predicted versus observed TVCs for the most efficient models—namely, MSI and FT-IR/MSI—and for the combined three-sensor model are illustrated in Figure 5. According to Figure 5A,B, the MSI and FT-IR/MSI models could estimate the TVCs within the acceptable area of ±1 log CFU/g, while an underestimation of the TVCs was evident for samples that exceeded 8 log CFU/g with the combined three-sensor model (Figure 5C).

The beta coefficients for the PLS-R models via MSI, FT-IR/MSI, and the three sensors in combination are provided in the linear equations given in Equations (1)–(3). Likewise, the contribution of each sensor to the estimation of the TVCs via the FT-IR/MSI model is presented in Equation (2). In Equation (2), the scores of the first six PCs (from the MSI data) and PC1 scores (from FT-IR) had values between 0.0991 to 14.8000 and were considered significant based on Marten’s uncertainty test. On the other hand, for the combined three-sensor model (Equation (3)), the b coefficients corresponding to the PC4 scores from the PCA analysis of the FT-IR data and to the PC1, PC3, and PC4 scores from the PCA analysis of the E-nose data demonstrated a significant contribution to the models’ development and prediction performance.
Y_TVCs/MSI_ = 13.984 + 9.529 × X_mean,405nm_ − 6.481 × X_mean,505nm_ + 13.632 × X_mean,570nm_ − 6.533 × X_mean,630nm_ + 5.323 × X_mean,645nm_ + 9.131 × X_mean,660nm_ − 8.421 × X_mean,700nm_ − 4.695 × X_mean,850nm_ + 4.261 × Χ_mean,890nm_ − 6.315× X_SD,405nm_ − 4.904 × X_SD,435nm_ + 9.588 × X_SD,470nm_ + 5.172 × X_SD,505nm_ + 3.452 × X_SD,525nm_ − 8.106 × X_SD,570nm_ − 3.473 × X_SD,850nm_ + 4.979 × X_SD,940nm_(1)
Y_TVCs/FT-IR/MSI_ = 7.374 − 1.606 × X_PC1/MSI_ − 3.963 × X_PC2/MSI_ + 2.914 × X_PC3/MSI_ + 2.794 × X_PC4/MSI_ + 7.930 × X_PC5/MSI_ + 14.800 × X_PC6/MSI_ − 0.091 × X_PC1/FT-IR_(2)
Y_TVCs/3 sensors_ =7.548 − 0.274 × X_PC4/FT-IR_ + 2.068 × X_PC1/E-nose_ − 6.20 ×X_PC3/E-nose_ + 2.198 ×X_PC4/E-nose_(3)

### 3.4. SVM-R Models for Assessing TVC Loads in Marinated Chicken Souvlaki

The values of the RMSE of the predictions of the developed SVM-R models for assessing the TVCs are presented in Table 2. Of the single-sensor models, MSI achieved the most efficient assessment of the TVCs, with RMSE values of cross-validation and prediction of 0.832 and 0.973 log CFU/g, respectively. Similarly, the combination of the PCA scores derived from the FT-IR and MSI data demonstrated that this was an acceptable linear SVM-R model, with an RMSE value of prediction that was close to 1.0 log CFU/g. On the contrary, the E-nose/FT-IR and MSI/E-nose models showed RMSE values of prediction that were over 1.5 log CFU/g; therefore, they were considered unacceptable. The same outcome was observed for the three-sensor model, where the RMSE of prediction was 1.938 log CFU/g. The individual E-nose and FT-IR models failed to accurately predict the TVCs, providing RMSE values that exceeded 1.921 log CFU/g.

The correlation between the predicted and observed TVCs derived from the SVM-R models developed on the MSI and FT-IR/MSI data is demonstrated in Figure 6 and Figure 7, respectively. In the case of MSI, the model optimization (Figure 6A) did not show differences between the observed and predicted TVCs, whereas there was a clear overestimation of TVCs between 4 and 6 log CFU/g during prediction (Figure 6B). In addition, for both the MSI and FT-IR/MSI models, an underestimation of the predicted TVCs occurred for samples with TVC loads of 8 log CFU/g (Figure 6B and Figure 7B). In addition, the SVM–regression beta coefficients for the MSI and FTI-IR/MSI models are documented in Figure 8 and Figure 9, respectively. Similarly to the PLS-R model with the implementation of MSI, the beta coefficients corresponding to the reflectance from 570 to 700 nm indicated their essential contribution in the prediction of spoilage in the chicken samples. Regarding the FT-IR/MSI model, the scores from PC1, PC2, PC5, and PC6 in the MSI analysis had a greater impact on TVC prediction, while the scores from PC1 from the FT-IR data analysis seemed to influence the models’ performance. In an attempt to improve the SVM-R model’s performance and find the appropriate kernel function, the Bayesian optimization process was employed, and the estimated combination of parameters and functions (with the minimum MSE) is provided in the Appendix A.

### 3.5. Classification Models for Assessing Spoilage in Marinated Chicken Souvlaki

The performance of the two-class models in terms of overall accuracy (%) is presented in Figure 10 for the individual and combined classification models. For the spectroscopic sensor models (MSI, FT-IR, FT-IR/MSI), the overall accuracy of their predictions exceeded 60% in most cases, with the highest percentages being obtained for FT-IR/MSI (LDA: 85%, LSVM: 82.5%, CSVM: 87.5%), followed by MSI with the SVM models (LSVM: 80% and QSVM: 77.5%). Concerning the two-class FT-IR models, the overall accuracy in their performance was recorded below 60%, which cannot be considered satisfactory. Likewise, the E-nose models could not exceed 39.4%, while the highest percentage of the overall accuracy of MSI/E-nose models was calculated at 48.48% for the LDA model. In contrast, the CSVM model developed on the combined FT-IR and E-nose data identified the correct class of the samples satisfactorily, with an overall accuracy of 63.63%. For the three-sensor models, the use of CSVM exhibited the most accurate discrimination between samples, with an overall accuracy of 72.73%. 

The performance of MSI (combined with the LSVM and CSVM models) is demonstrated in the confusion matrix (Table 3), where the per class sensitivity (%) and precision (%) are presented. For the MSI and LSVM model, 149 out of 169 samples and 32 out of 40 samples were accurately classified into their respective classes during model development and prediction, respectively. The sensitivity and precision for class 1 (non-spoiled samples) reached 90.14% and 83.12% for model development, respectively, while for model prediction, the respective percentages were 83.12% and 85.0%. The CSVM model developed on MSI data classified 127 out of 169 samples and 31 out of 40 samples into the correct classes for model development and prediction, respectively. For this model, the sensitivity for class 1 (non-spoiled) was calculated at 74.65% and 88.89% for cross-validation and prediction, respectively, whereas the precision ranged from 68.83 to 72.41%. In both MSI models, the sensitivity for class 2 (spoiled) was approximately the same as that for class 1 during cross-validation, with the exception of the prediction of the CSVM model, where one spoiled sample was misclassified as non-spoiled.

For the two-sensor and three-sensor models, the respective confusion matrix and the performance indexes of sensitivity and precision are presented in Table 4. For the LDA model that was developed on combined FT-IR/MSI data, 141 out of 169 samples and 34 out of 40 samples were correctly classified during model cross-validation and prediction, respectively. The calculated sensitivity and precision for class 1 amounted to 80% and 86.36%, respectively, for model cross-validation and prediction. The LSVM model correctly identified 143 out of 169 samples and 33 out of 40 samples during cross-validation and prediction, respectively. Moreover, the LSVM model developed on FT-IR/MSI data provided sensitivity and precision for class 1 of 87.14% and 78.20% for cross-validation, respectively, and 77.27% and 89.47% for prediction, respectively. For the CSVM model, 135 out of 169 samples and 35 out of 40 samples were classified into the correct classes during cross-validation and prediction, respectively. The sensitivity and precision for class 1 were 77.14% and 75% for model cross-validation, respectively, and 90% and 86.95% for model prediction, respectively. For the three-sensor model, 154 out of 169 samples and 29 out of 40 samples were properly classified during cross-validation and prediction, respectively. The sensitivity for class 1 reached 90.77% and 88.23% for cross-validation and prediction, while the respective values for precision were 86.76% and 68.18%. The sensitivity values for class 2 were, in most cases, similar to the sensitivity values for class 1, exceeding 80%, with the exception of the three-sensor model where the sensitivity for class 2 was calculated at 66.67%. This outcome indicated the potential of the developed models to accurately identify and categorize both non-spoiled (class 1) and spoiled (class 2) samples.

## 4. Discussion

So far, the spoilage, freshness, and safety of food products are evaluated using sensory and microbiological analyses [47]. The disadvantages of both approaches are that the first (sensory assessment) relies on highly trained panelists, which makes it costly and unattractive for routine analysis, while the second (microbiological analysis) provides retrospective results regardless of whether these have been analyzed with traditional methods (e.g., total viable counts) or with the use of molecular tools (e.g., PCR, RT-PCR, DGGE). It should be noted that these conventional methodologies are often misleading, and scientists have shown that it is more meaningful to measure the fraction of the microbiota contributing to spoilage [47], to use biomarkers [51], or to create libraries of fingerprints that can be elaborated with machine learning [52], thus offering a black-box approach that successfully addresses the issue of rapid responses to quality/freshness demands that are raised by the stakeholders (e.g., food business operators, consumers, and authorities).

The validated quantitative models confirmed the suitability of MSI and FT-IR sensors for the assessment of TVCs in marinated chicken souvlaki. Nevertheless, these non-invasive methods have been proposed in tandem with PLS-R models as rapid and efficient tools for the detection of spoilage/freshness in meat and poultry [53,54]. Regarding the E-nose, its performance was not in full agreement with the results of previous studies where this sensor in combination with PLS-R successfully predicted microbial populations in chicken stored in modified atmospheres [22] and the quality changes due to chicken fat oxidation [16]. This discrepancy could be attributed to the effect of the marination treatment performed on the poultry meat used in this study. The complexity of the surface on the chicken thigh muscle in combination with the presence of organic acids in the marinade could influence MOS signals in the E-nose analysis [18,39] and could, thus, explain the low performance of the models developed with the E-nose signals. The improved PLS-R model developed with the combined E-nose/MSI data confirmed that the use of HSI and/or MSI sensor data could improve the predictions of an E-nose for meat freshness assessments [32,34]. Indeed, the MSI and E-nose techniques in combination with NIR have been reported as a reliable and alternative approach to the estimation of total volatile basic nitrogen (TVB-N) in pork [28].

In general, the results obtained here indicated that the SVM-R models performed similarly to the PLS-R models, with the exception of the FT-IR model, in which the SVM–regression model presented higher RMSE values for k-fold cross-validation and prediction. The SVM-R model developed with MSI data exhibited the lowest RMSE values compared to those of all other single-sensor SVM-R models, whereas the individual E-nose and FT-IR models could not efficiently estimate the TVCs. Even though E-nose analysis provides a smaller data size than the other two spectroscopic methods [20], it seems that microbial spoilage can be described more thoroughly by the other two techniques. In a previous work, it was reported that the SVM Gaussian kernel function (RBF) combined with PLS-R and an E-nose was suggested as a feasible and rapid method for the estimation of pork microbiota [23]. Likewise, an SVM–regression model with an RBF kernel coupled with an E-nose was able to successfully predict the microorganisms involved in spoilage in beef meat [24].

From the performance indices of the classification models for the two quality classes, the obtained results indicated that the LSVM models developed with MSI and FT-IR/MSI data and the LDA models developed with FT-IR/MSI data presented good overall accuracy, sensitivity, and precision. The combination of the SVM models with the above-mentioned spectroscopic techniques, as well as the use of LDA with FT-IR data, was recommended in recent studies as an effective approach to the assessment of the quality of meat freshness [2]. The superiority of the LSVM model when coupled with MSI data has also been reported in similar studies for quality assessment in meat [12,23,24] and, specifically, in chicken thigh fillets [13]. The high overall accuracy of all FT-IR/MSI models confirmed that the combination of these nondestructive techniques could effectively more assess the quality of meat [28]. 

The findings reported in this work highlight the importance of choosing the appropriate machine learning model depending on the sensors’ features, as well as the necessity of combining data from different sensors in order to improve the performance of the developed models. Even though the quantitative and qualitative models developed with E-nose data could not accurately classify the samples into their respective quality classes, the fused model of the MSI/E-nose provided improved performance metrics. In contrast, MSI and the fusion of FT-IR and MSI data were proven to be effective for the assessment of the microbiological quality in marinated chicken souvlaki, regardless of the product batch, storage conditions, or analyst.

## 5. Conclusions

MSI, FT-IR, and an E-nose were evaluated for their potential in assessing the microbiological quality of marinated chicken souvlaki either as standalone sensors or via data fusion by using a variety of linear/nonlinear quantitative and qualitative models. For the assessment of TVCs via the PLS-R models, the MSI data provided the most accurate predictions, followed by the FT-IR/MSI model and the combined three-sensor model. Likewise, the SVM-R models developed with MSI and FT-IR/MSI data exhibited the most satisfactory estimation of the TVCs. For the classification of the samples into two quality classes, the LSVM models developed with MSI and FT-IR/MSI data and the LDA model developed with FT-IR/MSI data were able to provide correct classifications into the respective quality classes (non-spoiled vs. spoiled) with high overall correct classification percentages. The performance of the models was further confirmed through external validation by using data from independent poultry meat batches and different analysts. Overall, for the assessment of the microbiological quality of marinated chicken souvlaki, the implementation of spectroscopic methods, such as MSI and FT-IR, could provide encouraging results.

## Figures and Tables

**Figure 1 microorganisms-10-02251-f001:**
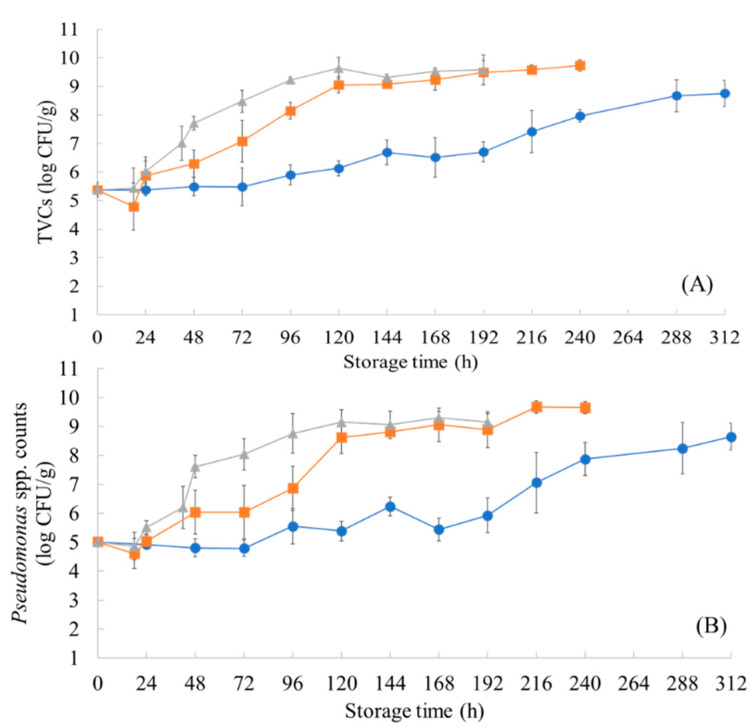
Changes in the population of TVCs (**A**) and *Pseudomonas* spp. (**B**) in marinated chicken souvlaki samples during storage at 10 (triangle), 5 (square), and 0 (circle) °C. The data points are mean values ± standard deviation (n = 4).

**Figure 2 microorganisms-10-02251-f002:**
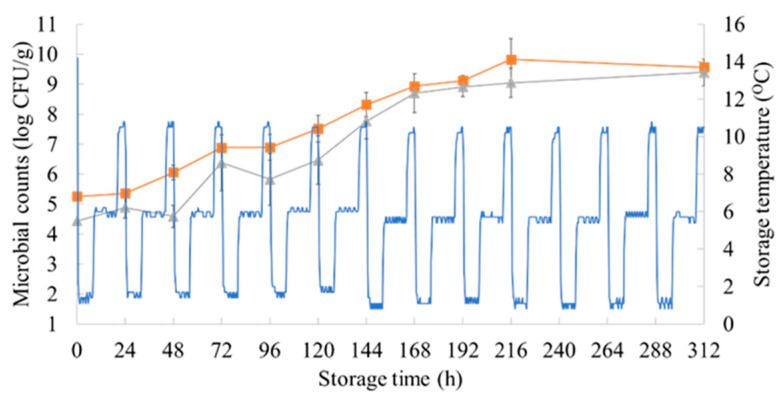
Changes in the population of TVCs (square) and *Pseudomonas* spp. (triangle) in marinated chicken souvlaki samples during storage with the dynamic temperature profile (blue line) (12 h at 0 °C, 8 h at 5 °C, and 4 h at 10 °C).

**Figure 3 microorganisms-10-02251-f003:**
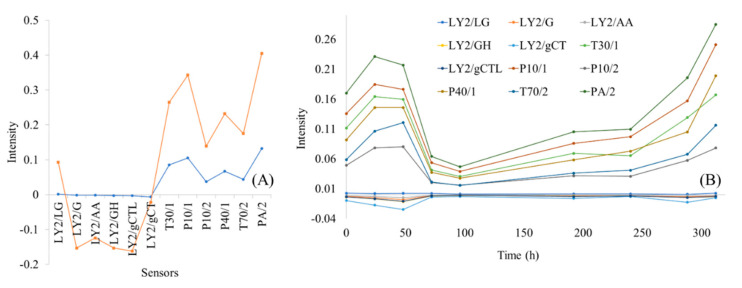
Signal (intensity) response from the E-nose instrument for non-spoiled (blue line: 0 h) and spoiled (orange line: 240 h at 5° C) marinated chicken souvlaki samples (**A**). Signal response from each E-nose sensor for spoiled samples (240 h at 5 °C) (**B**).

**Figure 4 microorganisms-10-02251-f004:**
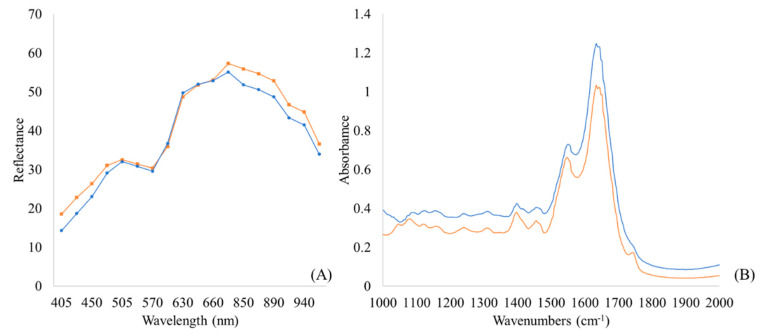
Typical MSI (405–970 nm) (**A**) and FT-IR spectra (1000–2000 cm^−1^) (**B**) for non-spoiled (blue line: 0 h) and spoiled (orange line: 240 h) marinated chicken souvlaki at 5 °C.

**Figure 5 microorganisms-10-02251-f005:**
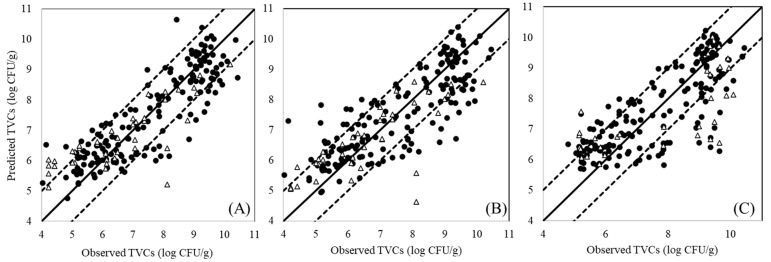
Predicted versus observed TVCs resulting from the development of the PLS-R model based on data from: MSI (**A**), FT-IR/MSI (**B**), and a combination of the three sensors (**C**) for cross-validation (solid symbols) and prediction (open symbols). The solid line represents the line of equity (y = x), while the dashed lines indicate the limit area of ±1.0 log CFU/g.

**Figure 6 microorganisms-10-02251-f006:**
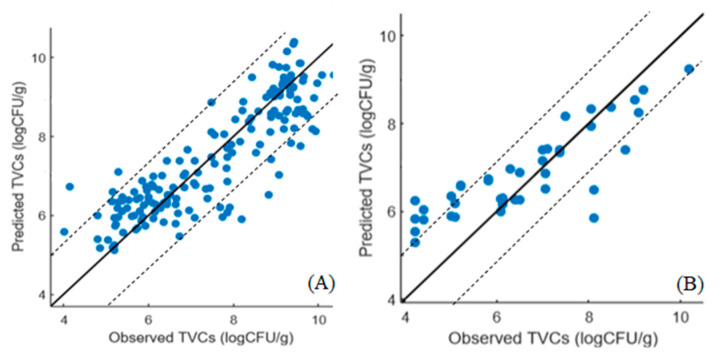
Predicted versus observed TVCs resulting from the SVM-R model with MSI data for cross-validation (**A**) and prediction (**B**). The solid line represents the line of equity (y = x).

**Figure 7 microorganisms-10-02251-f007:**
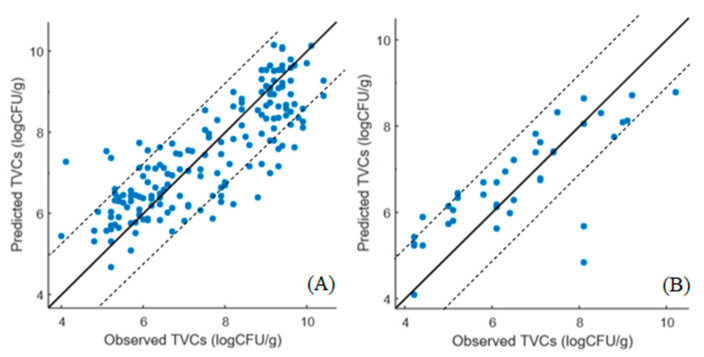
Predicted versus observed TVCs resulting from the SVM-R model with FT-IR/MSI data for cross-validation (**A**) and prediction (**B**). The solid line represents the line of equity (y = x).

**Figure 8 microorganisms-10-02251-f008:**
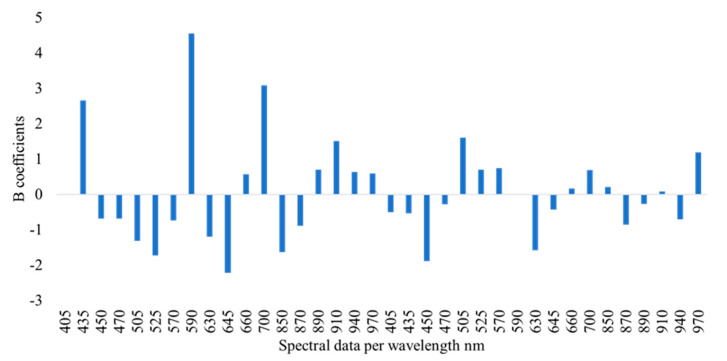
Beta (B) coefficients of the SVM-R model developed with MSI spectral data (mean intensity of pixels per wavelength) for marinated chicken souvlaki.

**Figure 9 microorganisms-10-02251-f009:**
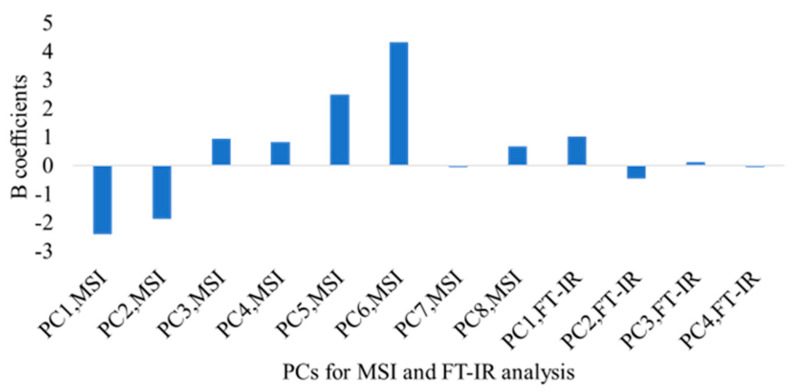
Beta (B) coefficients of the SVM-R model developed with FT-IR/MSI data (PCA scores) for marinated chicken souvlaki.

**Figure 10 microorganisms-10-02251-f010:**
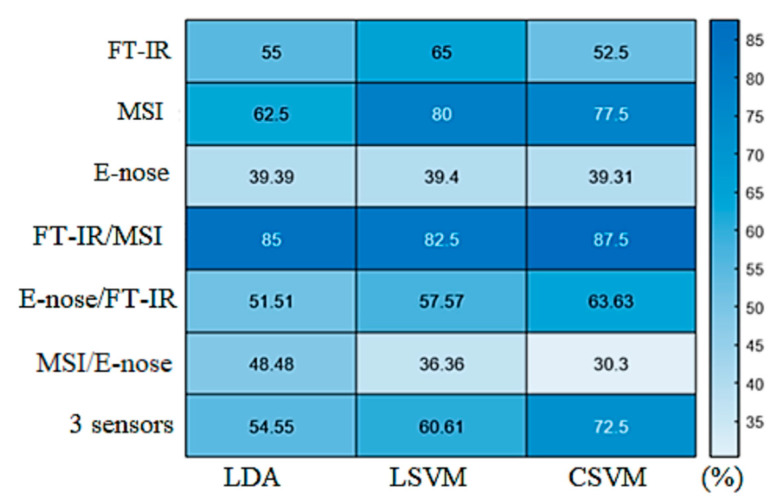
Heatmap presenting the performance (overall accuracy in %) of the LDA, LSVM, and CSVM models that were developed separately and in combination for each sensor for the classification of marinated chicken samples in two quality classes.

**Table 1 microorganisms-10-02251-t001:** PLS-R model parameters (slope, offset) and performance metrics (correlation coefficient, r; root mean squared error, RMSE) for the estimation of the TVCs in marinated chicken souvlaki samples via MSI, FT-IR, and E-nose analyses.

Sensor	Process	Observations	Slope	Offset	Correlation Coefficient, r	Root Mean Squared Error, RMSE (Log CFU/g)
**MSI**	FCV ^1^	169	0.776	1.698	0.868	0.815
	Prediction	40	0.511	3.419	0.803	0.998
**FT-IR**	FCV	169	0.62	2.87	0.746	1.099
	Prediction	40	0.374	4.902	0.497	1.627
**E-nose**	FCV	169	0.576	3.232	0.757	1.12
	Prediction	40	0.044	6.145	0.245	1.921
**MSI/FT-IR**	FCV	169	0.687	2.363	0.818	0.941
	Prediction	40	0.592	2.689	0.783	0.983
**FT-IR/E-nose**	FCV	169	0.598	3.055	0.758	1.131
	Prediction	40	0.171	6.245	0.222	1.757
**MSI/E-nose**	FCV	169	0.596	3.061	0.75	1.149
	Prediction	40	0.503	3.498	0.727	1.373
**Three sensors**	FCV	169	0.596	3.056	0.751	1.148
	Prediction	40	0.474	3.821	0.722	1.367

^1^ FCV: Full cross-validation.

**Table 2 microorganisms-10-02251-t002:** Performance of the SVM-R model (RMSE of cross-validation and prediction) when using MSI, FT-IR, and E-nose sensors (individually and in combination).

	Sensor					
	E-Nose		FT-IR		MSI	
**Step**	**k-CV ^1^**	**Prediction**	**k-CV**	**Prediction**	**k-CV**	**Prediction**
**RMSE (log CFU/g)**	1.311	1.921	1.846	3.583	0.832	0.973
	**E-nose/FT-IR**		**FT-IR/MSI**		**MSI/E-nose**	
**Step**	**k-CV**	**Prediction**	**k-CV**	**Prediction**	**k-CV**	**Prediction**
**RMSE (log CFU/g)**	1.06	1.579	0.953	0.999	1.134	1.658
	**3-sensors**					
**Step**	**k-CV**			**Prediction**		
**RMSE (log CFU/g)**	1.022			1.938		

^1^ k-CV: k-fold cross-validation.

**Table 3 microorganisms-10-02251-t003:** Confusion matrix and performance metrics of the developed models (LSVM, CSVM) for the classification of samples in two quality classes via MSI data.

Sensor	Model	Step	Confusion Matrix			Performance Metrics	
**MSI**	**LSVM**	**k-CV**	o/p	Class 1 ^1^	Class 2 ^2^	Sensitivity (%)	Precision (%)
			Class 1	64	7	90.14	83.12
			Class 2	13	85	86.75	
		**Prediction**	o/p	Class 1	Class 2	Sensitivity (%)	Precision (%)
			Class 1	17	5	77.27	85
			Class 2	3	15	83.33	
	**Model**	**Step**	**Confusion Matrix**			**Performance metrics**	
	**CSVM**	**k-CV**	o/p	Class 1	Class 2	Sensitivity (%)	Precision (%)
			Class 1	53	18	74.65	68.83
			Class 2	24	74	75.51	
		**Prediction**	o/p	Class 1	Class 2	Sensitivity (%)	Precision (%)
			Class 1	21	1	95.45	72.41
			Class 2	8	10	55.55	

^1^ Class 1: non-spoiled. ^2^ Class 2: spoiled.

**Table 4 microorganisms-10-02251-t004:** Confusion matrix and performance metrics of the developed models (LSVM, CSVM) for the classification of samples in two quality classes via FT-IR/MSI and three-sensor data.

Sensor	Model	Step	Confusion Matrix			Performance Metrics	
**FT-IR/MSI**	**LDA**	**k-CV**	o/p	Class 1	Class 2	Sensitivity (%)	Precision (%)
			Class 1	56	14	80	80
			Class 2	14	85	85.86	
		**Prediction**	o/p	Class 1	Class 2	Sensitivity (%)	Precision (%)
			Class 1	19	3	86.36	86.36
			Class 2	3	15	83.33	
	**Model**	**Step**	**Confusion Matrix**			**Performance metrics**	
	**LSVM**	**k-CV**	o/p	Class 1	Class 2	Sensitivity (%)	Precision (%)
			Class 1	61	9	87.14	78.20
			Class 2	17	82	82.83	
		**Prediction**	o/p	Class 1	Class 2	Sensitivity (%)	Precision (%)
			Class 1	17	5	77.27	89.47
			Class 2	2	16	88.89	
	**Model**	**Step**	**Confusion Matrix**			**Performance metrics**	
	**CSVM**	**k-CV**	o/p	Class 1	Class 2	Sensitivity (%)	Precision (%)
			Class 1	54	16	77.14	75
			Class 2	18	81	81.82	
		**Prediction**	o/p	Class 1	Class 2	Sensitivity (%)	Precision (%)
			Class 1	20	2	90	86.95
			Class 2	3	15	83.33	
**Three sensors**	**Model**	**Step**	**Confusion Matrix**			**Performance metrics**	
	**CSVM**		o/p	Class 1	Class 2	Sensitivity (%)	Precision (%)
		**k-CV**	Class 1	59	6	90.77	86.76
			Class 2	9	95	91.34	
		**Prediction**	o/p	Class 1	Class 2	Sensitivity (%)	Precision (%)
			Class 1	15	2	88.23	68.18
			Class 2	7	14	66.67	

## Data Availability

The datasets generated for this study are available on request from the corresponding author.

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
