# Peer review of "Assessment of the Microbial Spoilage and Quality of Marinated Chicken Souvlaki through Spectroscopic and Biomimetic Sensors and Data Fusion"

_microorganisms, 2022, doi:10.3390/microorganisms10112251_

Round 1
Reviewer 1 Report
This research compared different sensor and model combinations for non-destructive testing of the microbiological quality of marinated chicken souvlaki. Experiments and analysis results were conducted adequately and presented well. The results illustrated that FT-IR/MSI data analyzed by the CSVM algorithm provided better overall performance than other combinations. Although the accuracy, sensitivity, and precision of all combinations still need to be optimized, it is a useful work for determining optional rapid detection. Meanwhile, there remain a lot of minor grammar mistakes, especially the use of the article (the, a, an…) and the comma symbol.
Minor comments:
L24 and many lines the PLS-R
L29 the CSVM
L30 and many lines an overall
L37 such as
L39 the quality
L49 accurately predicted
L54 in -> into
L90 …, namely…a feasible…
L96 sensor
L104 the slaughter
L119 …, and afterward…
L153 sample
L154 non-useful
L185 Savitzky–Golay?
L202 and L207 the single-sensor model
L222 great -> significant
L275 and many lines the RMSE
L281 a very
L285 the underestimation
L297-300 Unclear, please rephrase
L303 a significant
L314-318 Unclear, please rephrase
L321 and many lines a RMSE
L325 The format of Table 2. should be checked.
L331 a clear
L337 important -> essential
L341 the Bayesian
L342 resulted -> estimated
L383 and many lines cross-validation
L460 also been
L469 accurately classify
Author Response
Reviewer 1
Comment 1: This research compared different sensor and model combinations for non-destructive testing of the microbiological quality of marinated chicken souvlaki. Experiments and analysis results were conducted adequately and presented well. The results illustrated that FT-IR/MSI data analyzed by the CSVM algorithm provided better overall performance than other combinations. Although the accuracy, sensitivity, and precision of all combinations still need to be optimized, it is a useful work for determining optional rapid detection
Response: We would like to thank the reviewer for his/her positive comments on our work.
Comment 2: Meanwhile, there remain a lot of minor grammar mistakes, especially the use of the article (the, a, an…) and the comma symbol.
Response: The minor grammar mistakes were detected and corrected throughout the manuscript.
Comment 3: L297-300 Unclear, please rephrase
Response: The manuscript was revised and the rephrased version is available in L 303-307.
Comment 4: L314-318 Unclear, please rephrase
Response: The manuscript was revised and the rephrased version is available in L 320-324.
Comment 5: L325 The format of Table 2. should be checked.
Response: The format of Table 2 was checked and modified.
Reviewer 2 Report
In this paper, the authors studied the microbial spoilage and quality of marinated chicken souvlaki through spectroscopic and biomimetic sensors and data fusion. I can clearly see the conceptual framework on which the authors have built their work (indeed very well explained throughout the paper). Still, I have problems seeing why they have used a marinated product with small consumption (worldwide). Although the methods of study are well-defined, there are some questions and minor changes that need to be clarified before considering its publication.

Author Response
Reviewer 2
Comment 1: In this paper, the authors studied the microbial spoilage and quality of marinated chicken souvlaki through spectroscopic and biomimetic sensors and data fusion. I can clearly see the conceptual framework on which the authors have built their work (indeed very well explained throughout the paper). Still, I have problems seeing why they have used a marinated product with small consumption (worldwide). Although the methods of study are well defined, there are some questions and minor changes that need to be clarified before considering its publication.
Response: We would like to thank the reviewer for his/her positive comments on our work.
Comment 2: Title: Typo, please, change the word "though" with "through".
Response: The title of the manuscript was revised.
Comment 3: Lines 75-79: The authors have correctly mentioned the complexity of the food matrix. Considering this comment, I would suggest the authors include a few lines about the complexity of testing a marinated product and how difficult or possible it could be to build a model predicting the microbiological and safety aspects of such a product?
Response: In order to illustrate the complexity of the examined food matrix a few lines were added in the manuscript (L446-449).
Comment 4: Lines 131-132: Could the authors please include the concentration of each compound in the chicken souvlaki marinate?
Response: The marinated samples of the presented experiments were provided by a Greek poultry industry and products marinade (as well as the concentration of each compound) is subjected to limitations imposed by the industry. However, they could be available upon request provided that we have the agreement from the industry.
Comment 5: Line 165: Please, mention on the FTIR analysis if the background absorption was subtracted.
Response: The reference with the procedure of the background measurement was added to the FT-IR analysis section, as well as the software with which the background absorption was subtracted (L168-170).
Comment 6: Lines 194-195: I understand this is a two levels classification. I would suggest the authors consider using the term "non-spoiled" or "good for consumption", etc. instead of the term "fresh".
Response: After consideration, the term "fresh" was replaced with the term "non-spoiled" through the manuscript.
Comment 7: Line 256: Regarding the FTIR and MSI spectra. What are the spectra of raw non-marinated chicken meat?
Response: The FT-IR and MSI spectra of the non-marinated chicken thigh fillets have been investigated (in a variety of storage temperature conditions and different batches) in previous studies in our laboratory. The most representative spectra for these two analyses could be found in reference 13 (Spyrelli, E.D., Papachristou, C., Nychas, G.J.E., Panagou, E.Z. Microbiological Quality Assessment of Chicken Thigh Fillets Using Spectroscopic Sensors and Multivariate Data Analysis. Foods 2021, 10, 2723. https://doi.org/10.3390/foods10112723).
Question 1: Why have the authors investigated the microbial spoilage and quality of marinated chicken souvlaki at 10 oC instead of the abused temperature of 8 oC?
Response: The storage temperature profiles were selected based on a published work on temperature changes in Greek refrigerators at household level (Vaikousi, H.; Biliaderis, C. G.; Koutsoumanis, K. P. Applicability of a microbial Time Temperature Indicator (TTI) for monitoring spoilage of modified atmosphere packed minced meat. Int. J. Food Microbiol. 2009, 133(3), 272-278. DOI: https://doi.org/10.1016/j.ijfoodmicro.2009.05.030).
Question 2: Why have the authors tested marinated chicken souvlaki?
Response: In previous studies, we had investigated FT-IR and MSI analysis efficacy (separately) for the quality assessment in chicken breast, chicken burger and chicken thigh fillets. After consideration, we selected this product based on the popularity of this type of food in Greece. Moreover, the Greek poultry industry, with which we were in collaboration during this project (QAPP), suggested this product based on its increased demand.
Question 3: Have the authors considered how the changes in the compounds' concentration in the marinate will affect the results of these sensors?
Response: Indeed, the changes in the compound’s concentration in the marinade could affect the results of the utilized sensors and thus models’ performance. However, the mixture of the marinade was fixed by the poultry company (stable recipe, homogenous pounder, certified supplier). Moreover, the marinade mixture pounder used for these experiments was from the same lot.